**Data Availability Statement:** Ethical restrictions exist (The Ethical Committee of the Local Health

# Exposure to low levels of hydrogen sulphide and its impact on chronic obstructive pulmonary disease and lung function in the geothermal area of Mt. Amiata in Italy: The cross-sectional InVETTA study

**Giorgia Stoppa** [1][⊕][¤a][*], **Daniela Nuvolone** [1][⊕], **Davide Petri** [1][¤b], **Letizia Centi** [2], **Francesca Nisticò** [2], **Emanuele Crocetti** [1], **Fabio Barbone** [3], **Fabio Voller** [1]

**1** Unit of Epidemiology, Regional Health Agency of Tuscany, Florence (FI), Italy, **2** Health Agency of South-East Tuscany, Arezzo (AR), Italy, **3** Department of Medicine, Surgery and Health Sciences, University of Trieste, Trieste, Italy

⊕ These authors contributed equally to this work.
¤a Current address: Unit of Biostatistics, Epidemiology and Public Health, Department of Cardiac, Thoracic, Vascular Sciences and Public Health, University of Padova, Padova (PD), Italy
¤b Current address: Department of Clinical and Experimental Medicine, University of Pisa, Pisa (PI), Italy
* giorgia.stoppa@phd.unipd.it

## Abstract

### Background

The geothermal power plants for electricity production currently active in Italy are all located in Mt. Amiata area in the Tuscany region. A cross-sectional survey was conducted in the framework of the regional project "InVETTA—Biomonitoring Survey and Epidemiological Evaluations for the Protection of Health in the Amiata Territories", using objective measures of lung function to investigate the role of hydrogen sulphide ($H_2S$) in affecting the respiratory health of the population living in this area.

### Methods

2018 adults aged 18–70 were enrolled during 2017–2019. Home and workplace addresses of participants were geocoded. Dispersion modelling was used to evaluate the spatial variability of exposure to $H_2S$ from the geothermal power plants' emissions. We estimated average long-term historical exposure to $H_2S$ and more recent exposure indicators. Chronic Obstructive Pulmonary Disease (COPD) was defined according to the Global Initiative for Chronic Obstructive Lung Disease (GOLD). Multivariable logistic regressions were performed to investigate associations between outcome and exposure.

### Results

Our findings did not showed any evidence of an association between increasing $H_2S$ exposure and lung function impairments. Some risk reductions were observed: a -32.8% (p =

Agency of South-East Tuscany, Italy) on sharing a de-identified data set because data contain potentially identifying and sensitive people's information (age, gender, residence, health status). A de-identified data set is not possible to provide due to ethical and legal considerations. Data are available from (contact: ausltoscanasudest@postacert.toscana.it, ars@postacert.toscana.it) for researchers who meet the criteria for access to confidential data.

**Funding:** This work was supported by regional funding provided by Tuscany Region – Direzione Generale Politiche Ambientali, Energia e Cambiamenti Climatici (Resolution N. 973 on November 10, 2014)(Grant recipient: Dr. Fabio Voller). The funding source had no role in the study design, data collection, analysis and interpretation of data, the writing of the report, and the decision to submit the article for publication.

**Competing interests:** The authors have declared that no competing interests exist.

**Abbreviations:** InVETTA, Biomonitoring Survey and Epidemiological Evaluations for the Protection of Health in the Amiata Territories; COPD, Chronic Obstructive Pulmonary Disease; GOLD, Global Initiative for Chronic Obstructive Lung Disease; ARPAT, Regional Agency for the Environmental Protection of Tuscany; $CO_2$, carbon dioxide; $CH_4$, methane; $H_2S$, hydrogen sulphide; WHO, World Health Organization; ATSDR, Agency for Toxic Substances and Disease Registry; FVC, forced vital capacity; FEV1, forced expiratory volume in one second; ERS, European Respiratory Society; WRF, Weather Research and Forecasting Model; $H_2S_{past}$, maximum average mobile concentration of $H_2S$ calculated over 90 days according to the past emissions scenario; $H_2S_{current}$, maximum average mobile concentration of $H_2S$ calculated over 90 days according to the current emissions scenario; AMIS, filters for the abatement of hydrogen sulphide and mercury; SD, standard deviation; GIS, geographical information system; PR, prevalence ratio; BMI, body mass index; PM, particulate matter; $NO_2$, nitrogen oxides.

0.003) for FEV1<80% and a -51.7% (p = 0.001) risk decrease for FVC<80% were associated with interquartile increase (13.8 µg/m³) of $H_2S$ levels.

## Conclusion

Our study provides no evidence that chronic exposure to low levels of $H_2S$ is associated with decrements in pulmonary function, suggesting that ambient $H_2S$ exposure may benefit lung function.

## Introduction

The advent of industrial geothermal energy utilization in Italy dates back to the 1900s. The first geothermoelectric power plant in the world was built in 1913 in Italy, in the province of Pisa. Today, in Italy, geothermal power plants are only found in Tuscany, in the two geothermal districts of Larderello-Travale and Mt. Amiata. The over 6 billion KWh produced by the 36 geothermal electric plants satisfy more than 33% of the region's energy needs, thus giving a significant contribution to the electricity balance from renewable sources in Tuscany [1, 2].

Geothermal energy is a renewable energy source consisting of a body of hot aqueous fluid and hot rock continuously produced inside the earth's crust. This energy is extracted as steam or hot water from the surface and is used to heat and cool buildings (above 20° C/Km) or generate electricity [3–6].

However, geothermal energy exploitation has some adverse environmental effects, such as potential interaction with hydrological resources, subsidence problems, potential interference with natural seismicity and landscape damage. Furthermore, the atmospheric emission of non-condensable gases remains a major environmental issue related to the use of geothermal fluids to generate electricity.

In Tuscany, the geothermal power stations release pollutants such as carbon dioxide ($CO_2$), methane ($CH_4$) and hydrogen sulphide ($H_2S$). In addition, lower levels of several potentially harmful substances such as nitrogen, hydrogen, ammonia, boric acid, radon, rare gases and volatile trace elements (such as mercury, arsenic and antimony) are emitted [7–11]. As the main constituent of geothermal fluids, $H_2S$ represents a leading environmental concern for air, soil, and vegetation pollution. $H_2S$ is a flammable, colourless water-soluble gas with a characteristic odour of rotten eggs that is detectable at concentrations as low as 7 µg/m³. The health effects of exposure to high levels of $H_2S$ are well known, from respiratory, eye and throat irritation (5–29 mg/m³), to olfactory nerve paralyses (>140 mg/m³) and coma (>700 mg/m³) [12–14]. Conversely, the health effects associated with exposure to low levels of $H_2S$ in communities near natural or anthropogenic sources are less clear [15, 16]. The literature suggests a cardioprotective role for $H_2S$ in cardiac arrhythmias, cardiac fibrosis, heart failure, cardiac hypertrophy, ischaemia–reperfusion injury and myocardial infarction [17]. The WHO guidelines for protecting human health set tolerable concentration at the average value of 150 µg/m³ over 24 h, 100 µg/m³ over 14 days and 20 µg/m³ over 90 days [13, 14].

According to the Tuscany Region health databases, several descriptive and analytical epidemiologic studies have been conducted on the population living in the geothermal district of Mt. Amiata. Local mortality and hospitalization rates were compared with average regional rates reporting some risk excesses for respiratory diseases [18–21]. In particular, in a cohort of 33,804 residents in the Amiata area, observed for a total of 391,002 person-years, exposure to $H_2S$ ($H_2S$ indicator ranged from 0.5 to 33.5 µg/m) was associated with higher mortality (years

1998–2014) and hospital discharge rates (1998–2016) for respiratory diseases, chronic obstructive pulmonary disease (COPD) and disorders of the peripheral nervous system [20].

A cross-sectional survey was conducted in the framework of the regional project "InVETTA—Biomonitoring Survey and Epidemiological Evaluations for the Protection of Health in the Amiata Territories", using objective measures of lung function to investigate further the role of $H_2S$ in determining the observed respiratory morbidity in populations living in the Mt. Amiata area.

## Materials and methods

### Study design and participants

InVETTA was a cross-sectional study enrolling participants living in the geothermal area of Mt. Amiata. Participants' recruitment started on May 19th 2017, and ended on February 19th 2019.

The sample size was calculated using data from a prior monitoring survey that was carried out towards the end of the 1990s to examine the levels of mercury and arsenic in 900 respondents' blood and urine samples. The number of patients needed to detect a difference in arsenic and mercury concentrations was 160 and almost 1500, respectively, based on the assumption that the study would have a 90% power and a 0.05 alpha error. In order to ensure the survey's proper operation and to allow as many individuals as possible the chance to participate in the survey, the expected sample size was finally determined to be 2000 subjects in light of the extensive interest of the local public opinion on environmental hazards.

A comprehensive communication plan (traditional local media, public meetings, a website dedicated to the project, and social networks) was implemented to enhance local community participation. Selection criteria were: being 18–70 years old at the time of enrolment, possessing a phone number, and having lived in the Amiata area for at least five years. Residence was classified as follows: the "Main Area" is the one which is most affected by emissions from geothermal plants and includes the municipalities of Abbadia San Salvatore, Piancastagnaio, Arcidosso, Santa Fiora, Castel del Piano and Castell'Azzara, whereas the "Control Area" includes Seggiano, Radicofani, Cinigiano and Castiglione d'Orcia. Regarding geothermal plants emissions, the distinction between "Main area" and "Control area" was based on the air H2S concentrations estimated using dispersion models output. The validity of dispersion models was tested against data from air quality monitoring stations present in the area."

Initially, stratified random sampling from municipal registries was applied to obtain a representative sample of subjects by gender, age group (18–39, 40–54 and 55–70 years), municipality of residence, and exposure to geothermal emissions. Volunteers' participation was also encouraged, and the same selection criteria were applied to volunteers as for the random sampling. Trained health personnel contacted all participants, both sample and volunteers, to schedule the appointments for the spirometry and the administration of questionnaires. A face-to-face interview was administered by suitably trained staff to each participant about general information, lifestyle, medical history, and occupational exposure to improve $H_2S$ exposure evaluation and control for confounding. The same operator conducted more than 90% of the interviews to reduce the risk of introducing interviewer-related confounding. All participants' data were coded by assigning unique identifiers.

### Spirometric testing

Trained health professionals performed spirometry using the KoKo® Legend II nSpireHealth, Inc. spirometer following American Thoracic Society/European Respiratory Society performance guidelines [22]. The spirometry parameters to evaluate lung function included forced

vital capacity (FVC) and forced expiratory volume in one second (FEV1). Briefly, FVC is the maximum amount of air that can be exhaled when blowing out as fast as possible at the end of the manoeuvre; FEV1 is the air volume exhaled in the first second of the manoeuvre. The ratio of the two measurements (FEV1/FVC) was calculated. Spirometric values were compared with expected values calculated with the equations recommended by the Global Lung Function Initiative, ERS Task Force [23]. The prediction equations require age, gender and height. FEV1 and FVC are expressed as percentages of predicted values. COPD was defined according to the Global Initiative for Chronic Obstructive Lung Disease–GOLD [24] and the review by Halbert et al. [25] as follows:

- COPD Stage I: FEV1/FVC$<$70%

- COPD Stage II: FEV1/FVC$<$70% and the percent of the predicted value of FEV1$<$80%.

  The following were considered as additional indicators of reduced respiratory function:

- the percent of the predicted of FEV1$<$80% (FEV1$<$80%),

- the percentage of the predicted of FVC$<$80% (FVC $<$80%).

## Exposure estimation

Seven geothermoelectric power plants were activated in the Mt. Amiata area in the last decades. Table 1 summarizes the characteristics of these plants. The oldest and most polluting two, Bellavista and Piancastagnaio2 (PC2) were closed in 2000 and 2011, respectively, whereas the most recent, Bagnore 4, was constructed in 2014. Filters for abating hydrogen sulphide and mercury (AMIS technology) were progressively installed over the years.

H$_2$S air concentrations are constantly monitored by ENEL Green Power and the Regional Agency for the Environmental Protection of Tuscany (ARPAT). From 1997 to 2014, six fixed monitoring sites were installed to record H$_2$S hourly concentrations.

Dispersion modelling was used to evaluate the spatial variability of exposure to H$_2$S, whose technical and methodological details are described in Nuvolone 2019 [20]. In brief, an integrated framework using WRF, CALMET and CALPUFF modelling was implemented. CALMET provided 3D reconstructions of wind and temperature fields starting from meteorological measurements, orography and land use data. In this application, CALMET was used to downscale the wind fields produced by WRF (Weather Research and Forecasting Model), one of the most advanced mesoscale numerical weather prediction systems. CALPUFF was applied to assess H$_2$S long-range transport. To evaluate average long-term historical exposure to H$_2$S ("past scenario"), the emissions of six geothermal plants operating in the absence of any abatement filters (AMIS) were used as input data (PC2, Bellavista, PC3, PC4,

**Table 1. Characteristics of the geothermoelectric power plants in the Mt. Amiata area.**

| Power plant | Municipality | Power (MW) | Start | AMIS[a] Date of installation |
|---|---|---|---|---|
| Piancastagnaio 2 (PC2) | Piancastagnaio | 20 | 1968 closed in 2011 | Not installed |
| Bellavista | Piancastagnaio | 20 | 1987 closed in 2000 | Not installed |
| Piancastagnaio 3 (PC3) | Piancastagnaio | 20 | 04/05/1990 | 09/06/2006 |
| Piancastagnaio 4 (PC4) | Piancastagnaio | 20 | 28/11/1991 | 23/10/2008 |
| Piancastagnaio 5 (PC5) | Piancastagnaio | 20 | 02/02/1996 | 09/06/2006 |
| Bagnore 3 | Santa Fiora | 20 | 17/12/1998 | 08/02/2002 |
| Bagnore 4 | Santa Fiora | 40 | 31/11/2014 | 31/11/2014 |

[a] AMIS: Filters for the abatement of H$_2$S and mercury

PC5 and Bagnore3 plants). The maximum average mobile concentration of $H_2S$ calculated over 90 days ($H_2S_{past}$), i.e. the maximum value derived from the series of moving averages calculated over 90 days throughout the year, was selected as the most representative indicator of long-term $H_2S$ exposure, in line with WHO guidelines.

In addition, we considered a "current emissions scenario" ($H_2S_{current}$) which used emissions from the five plants currently active in the Mt. Amiata area (PC3, PC4, PC5, Bagnore 3 and Bagnore 4), excluding the oldest ones (Bellavista and PC2).

Geographical coordinates were assigned to each home address and workplace, and all subjects were included in a geographic information system (GIS). By overlapping with orthophotos and regional technical maps, the georeferencing results were subjected to data quality analysis, verifying the degree of completeness and precision of the geocoding process.

Exposure was assigned at an individual level considering residential history, exposure levels at home and work/school locations and time spent at home and work/school. In the case of multiple residence locations, we calculated a time-weighted average exposure metric, adding for each residence location the $H_2S$ air concentration multiplied by the time spent in each residence location and dividing by the total residence time. In addition, specific weights were introduced to account for exposure levels at home and work/school locations and time spent at work/school, distinguishing between full-time and part-time activities.

As for sensitivity analyses, we considered other exposure indicators, such as the aerial distance of home address from the nearest geothermal power plant, and the odour perception obtained from the questionnaire ("Do you habitually experience the classic 'rotten egg' smell of hydrogen sulphide?").

## Data analysis

Descriptive analyses were performed and the results are reported as frequency and percentage for categorical variables and mean and standard deviation (SD) for continuous variables. In addition, Wilcoxon-type tests were performed for the continuous variables, and the Pearson chi-square test was carried out for the categorical variables.

We fitted multivariable logistic regression for dichotomic outcomes: the estimated coefficients were expressed as Prevalence Ratio (PR) and 95% confidence intervals associated with interquartile range increase in $H_2S$ concentrations [26]. In addition, sensitivity analyses were performed, introducing categorical variables for $H_2S$ exposure metrics, using the tertiles of the distribution. For the continuous outcomes, multivariable analyses using generalized linear models were specified using a Gamma response with a log link, to reduce the skewness of spirometric measurements and provide a natural interpretation of regression coefficients in terms of percent change [27, 28]. Known risk factors for respiratory symptoms and lung function impairment were included a priori in the multivariate models: gender, age classes (18–39, 40–54, 55–70 years), education (low, medium, high), Body Mass Index (BMI) defined as kg/height$^2$, physical activity (active, partial active, sedentary), pack-year (0, <15, 15–30, >30) and cardiac comorbidities. Covariates examined for entry into regression models included mode of participation in InVETTA (sample vs volunteers), residence in main and control municipalities, passive smoking exposure, alcohol consumption, home dampness/mould, wood heating, pets, occupational exposure to dust, radiation, chemicals, working in a geothermal power plant, past working in mines. Additional analyses included stratified models considering gender, age group, mode of participation, smoking habits and residence in main and control areas. Missing values were excluded from the analyses.

All the analyses were performed using the STATA software version 16 (StataCorp LP, College Station, TX, USA).

## Results

In the InVETTA survey, spirometry was performed on 2018 out of 2060 participants (98%). Unfortunately, a minority of participants failed to complete the test for various reasons related to specific psychophysical conditions. Fifty-two percent of the total subjects were randomly extracted, and 978 were volunteers. Among non-volunteers, the participation rate was 41.6%. No significant differences in gender, age and average exposure to $H_2S$ from geothermal emissions were reported between respondents and non-respondents. Conversely, volunteers and randomly selected subjects reported some differences, as shown in Table 2. Volunteers lived in the main municipalities more frequently than the random sample, had a higher education level, showed a lower percentage of current smokers, a lower frequency of overweight/obesity and were less sedentary.

The $H_2S_{past}$ indicator ranged from 0.5 to 45.5 $\mu g/m^3$, with a mean of 10.4 $\mu g/m^3$, a standard deviation of 10.5 $\mu g/m^3$ and an interquartile range (IQR) of 13.8 $\mu g/m^3$. The $H_2S_{current}$ indicator showed lower values, ranging from 0.5 to 3.9 $\mu g/m^3$ with a mean of 0.8 $\mu g/m^3$, a standard deviation of 0.7 $\mu g/m^3$ and an IQR of 3.5 $\mu g/m^3$. Sixteen percent of the participants were historically exposed to $H_2S$ levels greater than the WHO guideline for medium/long-term exposure (20 $\mu g/m^3$) [13, 14]. On average, InVETTA participants resided 4.5 km from the nearest geothermal power plant, ranging from 64 m to more than 23 km. About 70% of the InVETTA

**Table 2. Comparison between volunteers and randomly selected subjects in the InVETTA survey.**

| | Total<br>N = 2018 | Random Sample<br>N = 1040 | Volunteers<br>N = 978 | p-values[a] |
|---|---|---|---|---|
| **Gender** | | | | |
| Men | 881 (43.7) | 472 (45.4) | 409 (41.8) | |
| Women | 1137 (56.3) | 568 (54.6) | 569 (58.2) | 0.107 |
| **Age (years)** | | | | |
| 18–39 | 526 (26.1) | 295 (28.4) | 231 (23.6) | 0.053 |
| 40–54 | 706 (34.9) | 352 (33.8) | 354 (36.2) | |
| 55–70 | 786 (39.0) | 393 (37.8) | 393 (40.2) | |
| **Residence** | | | | <0.001 |
| Main area | 1773 (88.1) | 852 (81.9) | 921 (94.2) | |
| Control area | 245 (11.9) | 188 (18.1) | 57 (5.8) | |
| **Education** | | | | |
| Low | 555 (27.5) | 354 (34.1) | 201 (20.6) | <0.001 |
| Medium | 993 (49.3) | 506 (48.7) | 487 (49.9) | |
| High | 468 (23.2) | 179 (17.2) | 289 (29.5) | |
| **Smoking** | | | | |
| Never smokers | 904 (45.2) | 467 (45.3) | 437 (45.1) | <0.001 |
| Ex-smokers | 610 (30.5) | 281 (27.2) | 329 (34.0) | |
| Smokers | 486 (24.3) | 284 (27.5) | 202 (20.9) | |
| **BMI** | | | | |
| Normal | 1066 (52.9) | 499 (48.0) | 567 (58.0) | <0.001 |
| Overweight/obese | 952 (47.1) | 541 (52.0) | 411 (42.0) | |
| **Physical activity** | | | | |
| Active | 507 (25.2) | 263 (25.3) | 244 (25.0) | <0.001 |
| Partial active | 732 (36.3) | 330 (31.8) | 402 (41.2) | |
| Sedentary | 776 (38.5) | 445 (42.9) | 331 (33.8) | |

[a] p-values from $\chi^2$ test

participants claimed to perceive the smell of $H_2S$. Of these, 45% reported perceiving it a few times a month, 16% every day and 10% rarely.

Descriptive results of respiratory function included a median FEV1 of 103% (SD = 12.8), a median FVC of 105% (SD = 12.5) and a mean FEV1/FVC of 78.0% (SD = 7.5). Comparing lung function by gender and age, we found a lower breathing performance in men aged 55–70. A total of 255 subjects, i.e. 12.6% of participants, reported COPD Stage I (FEV1/FVC<70%). Of these, 63 participants (3.1% of the total) reported COPD Stage II, and 7 subjects fell into the category 'severe' (FEV1/FVC<70% and 30%≤FEV1<50%). Six percent of participants, i.e. 127 subjects, reported a FEV1<80%, and 3.6%, i.e. 73 subjects, had a FVC<80%.

Table 3 reports the prevalence values of COPD and lung function indicators stratified by categories. Men showed a higher prevalence than women for COPD Stage I, whereas no difference by gender was observed for the other outcomes. Higher values were observed in the elderly, smokers, subjects with lower levels of education, subjects living in the control area, those with higher BMI, sedentary people, subjects with cardiovascular comorbidities and those who reported past working in mines.

Descriptive statistics of $H_2S$ indicators for both emissions scenarios stratified by outcomes are reported in Table 4. No significant differences in $H_2S$ average concentrations were observed between subjects diagnosed with COPD, Stage I or Stage II, whereas lower $H_2S$ levels were observed in those reporting FEV1 and FVC <80%.

Table 5 shows results from generalized linear models using lung function parameters as continuous dependent variables. Percent variation and 95% confidence intervals associated with the interquartile increase of $H_2S$ concentrations are reported, adjusting for confounding variables (gender, age, education, BMI, pack-years, physical activity, mode of participation in InVETTA and cardiac comorbidities). Past and current emissions scenarios exposure metrics were evaluated.

As for FEV1 and FVC parameters, slight increases were associated with increasing $H_2S$ concentrations, +1.10% and +1.66%, respectively. On the other hand, FEV1/FVC ratio showed a very slight decrease of -0.51%, associated with increased $H_2S$ exposure.

Table 6 shows the multivariate logistic regression results for COPD and lung function parameters using past and current emissions scenarios. Results from models using home distance from the nearest geothermal plant and odour perception are also shown. In both emissions scenarios, there was no evidence of an association between COPD and other lung function parameters and increasing $H_2S$ exposure. For all parameters analysed, except for COPD Stage I, increasing $H_2S$ exposure was negatively associated, and risk reductions were observed. In particular, a -32.8% (p = 0.003) risk reduction for FEV1<80% and a -51.7% (p = 0.001) risk decrease for FVC<80% were reported, in association with interquartile range increases in average historical $H_2S$ concentrations. Analyses stratified by gender, age, residence, mode of participation and smoking habits confirm results observed for all participants. The highest risk reductions for FEV1<80% were observed in women, smokers and volunteers. Living near a geothermal plant or perceiving $H_2S$ odour were not associated with increased risk for lung function. A reduced risk for COPD was reported for subjects living near a geothermal power plant.

## Discussion

In the InVETTA cross-sectional study, exposure to low levels (WHO guideline [13, 14]) of $H_2S$, in the range 0–45 μg/m$^3$, was not associated with chronic airway obstruction (COPD Stage I and Stage II) or any alteration of lung function. Results were confirmed in stratified analyses by gender, age group, smoking habits, residence area and mode of participation.

**Table 3. Prevalence of COPD and lung function outcomes (%) by categories.**

| | COPD Stage I | | | COPD Stage II | | | FEV1<80% | | | FVC<80% | | |
|---|---|---|---|---|---|---|---|---|---|---|---|---|
| | N | % | p[a] | N | % | p[a] | N | % | p[a] | N | % | p[a] |
| **Total** | 255 | 12.6 | | 63 | 3.1 | | 127 | 6.3 | | 73 | 3.6 | |
| **Gender** | | | | | | | | | | | | |
| Men | 135 | 15.3 | | 28 | 3.2 | | 50 | 5.7 | | 29 | 3.3 | |
| Women | 120 | 10.6 | <0.001 | 35 | 3.1 | 0.898 | 77 | 6.8 | 0.314 | 44 | 3.9 | 0.490 |
| **Age group (years)** | | | | | | | | | | | | |
| 18–39 | 19 | 3.6 | | 5 | 1.0 | | 15 | 2.9 | | 8 | 1.5 | |
| 40–54 | 69 | 9.8 | | 7 | 1.0 | | 29 | 4.1 | | 23 | 3.3 | |
| 55–70 | 167 | 21.3 | <0.001 | 51 | 6.5 | <0.001 | 83 | 10.6 | <0.001 | 42 | 5.3 | <0.001 |
| **Residence** | | | | | | | | | | | | |
| Main area | 215 | 12.1 | | 47 | 2.7 | | 98 | 5.5 | | 58 | 3.3 | |
| Control area | 40 | 16.3 | 0.064 | 16 | 6.5 | 0.001 | 29 | 11.8 | <0.001 | 15 | 6.1 | 0.025 |
| **Mode of participation** | | | | | | | | | | | | |
| Sample | 132 | 12.7 | | 40 | 3.9 | | 72 | 6.9 | | 35 | 3.4 | |
| Volunteers | 123 | 12.6 | 0.938 | 23 | 2.4 | 0.054 | 55 | 5.6 | 0.230 | 38 | 3.9 | 0.532 |
| **Pack-year** | | | | | | | | | | | | |
| 0 | 79 | 8.7 | | 14 | 1.6 | | 38 | 4.2 | | 27 | 3.0 | |
| <15 | 68 | 10.4 | | 19 | 2.9 | | 38 | 5.8 | | 21 | 3.2 | |
| 15–29 | 54 | 21.0 | | 9 | 3.5 | | 21 | 8.2 | | 12 | 4.7 | |
| >30 | 52 | 30.8 | <0.001 | 20 | 11.8 | <0.001 | 29 | 17.2 | <0.001 | 12 | 7.1 | 0.045 |
| **Education** | | | | | | | | | | | | |
| Low | 86 | 15.5 | | 28 | 5.1 | | 51 | 9.2 | | 33 | 6.0 | |
| Medium | 117 | 11.8 | | 23 | 2.3 | | 53 | 5.3 | | 27 | 2.7 | |
| High | 52 | 11.1 | 0.057 | 12 | 2.6 | 0.009 | 23 | 4.9 | 0.004 | 13 | 2.8 | 0.003 |
| **BMI** | | | | | | | | | | | | |
| Normal | 129 | 12.1 | | 27 | 2.5 | | 55 | 5.2 | | 31 | 2.9 | |
| Overweight/obesity | 126 | 13.2 | 0.444 | 36 | 3.8 | 0.107 | 72 | 7.6 | 0.026 | 42 | 4.4 | 0.071 |
| **Passive smoking** | | | | | | | | | | | | |
| No | 227 | 13.0 | | 56 | 3.2 | | 108 | 6.2 | | 62 | 3.5 | |
| Yes | 28 | 11.0 | 0.374 | 7 | 2.8 | 0.698 | 18 | 7.1 | 0.584 | 10 | 3.9 | 0.760 |
| **Alcohol** | | | | | | | | | | | | |
| Abstainer | 82 | 12.6 | | 24 | 3.7 | | 51 | 7.9 | | 29 | 4.5 | |
| Moderate | 144 | 12.9 | | 32 | 2.9 | | 59 | 5.3 | | 34 | 3.0 | |
| High risk | 23 | 12.4 | 0.982 | 5 | 2.7 | 0.585 | 12 | 6.5 | 0.095 | 6 | 3.2 | 0.284 |
| **Physical activity** | | | | | | | | | | | | |
| Active | 72 | 14.2 | | 16 | 3.2 | | 23 | 4.5 | | 13 | 2.6 | |
| Partial active | 77 | 10.5 | | 11 | 1.5 | | 36 | 4.9 | | 23 | 3.1 | |
| Sedentary | 105 | 13.5 | 0.097 | 35 | 4.5 | 0.003 | 67 | 8.6 | 0.002 | 37 | 4.8 | 0.081 |
| **Wood heating** | | | | | | | | | | | | |
| No | 107 | 12.7 | | 27 | 3.2 | | 54 | 6.4 | | 33 | 3.9 | |
| Yes | 148 | 12.7 | 1.000 | 36 | 3.1 | 0.884 | 73 | 6.3 | 0.895 | 40 | 3.4 | 0.567 |
| **Home dampness/mould** | | | | | | | | | | | | |
| No | 204 | 13.2 | | 47 | 3.0 | | 99 | 6.4 | | 57 | 3.7 | |
| Yes | 50 | 10.9 | 0.203 | 16 | 3.5 | 0.620 | 27 | 5.9 | 0.701 | 15 | 3.3 | 0.682 |
| Pets | | | | | | | | | | | | |
| No | 140 | 14.5 | | 28 | 2.9 | | 60 | 6.2 | | 44 | 4.6 | |
| Yes | 115 | 11.0 | 0.016 | 35 | 3.3 | 0.581 | 67 | 6.4 | 0.885 | 29 | 2.8 | 0.031 |

*(Continued)*

**Table 3.** (Continued)

| | COPD Stage I | | | COPD Stage II | | | FEV1<80% | | | FVC<80% | | |
|---|---|---|---|---|---|---|---|---|---|---|---|---|
| | N | % | p[a] | N | % | p[a] | N | % | p[a] | N | % | p[a] |
| **Occupational exposure (dust, radiation, chemicals)** | | | | | | | | | | | | |
| No | 136 | 11.6 | | 37 | 3.2 | | 74 | 6.3 | | 40 | 3.4 | |
| Yes | 119 | 14.1 | 0.097 | 26 | 3.1 | 0.921 | 53 | 6.3 | 0.973 | 33 | 3.9 | 0.557 |
| **Working in a geothermal power plant** | | | | | | | | | | | | |
| No | 250 | 12.7 | | 63 | 3.2 | | 126 | 6.4 | | 72 | 3.7 | |
| Yes | 5 | 9.8 | 0.537 | 0 | 0.0 | 0.194 | 1 | 2.0 | 0.197 | 1 | 2.0 | 0.521 |
| **Past working in mines** | | | | | | | | | | | | |
| No | 177 | 11.0 | | 47 | 2.9 | | 92 | 5.7 | | 52 | 3.2 | |
| Yes | 78 | 19.8 | <0.001 | 16 | 4.1 | 0.242 | 35 | 8.9 | 0.020 | 21 | 5.3 | 0.045 |
| **Cardiac comorbidities** | | | | | | | | | | | | |
| No | 198 | 11.7 | | 45 | 2.7 | | 84 | 5.0 | | 47 | 2.8 | |
| Sì | 57 | 17.2 | 0.007 | 18 | 5.4 | 0.008 | 43 | 13.0 | <0.001 | 26 | 7.8 | <0.001 |

[a]p: p-values from $\chi^2$ test

Estimating spatial variability of $H_2S$ air concentrations from dispersion modelling included both past and current emission settings. $H_2S$ concentrations in the Amiata area have sharply declined over the years because of the installation of $H_2S$ and mercury abatement filters and thanks to the closure of the most polluting power plants. Average $H_2S$ concentrations estimated for each participant in the InVETTA survey decreased from 10.4 μg/m³ to 0.8 μg/m³ when considering past and current emissions scenarios, both lower than WHO long-term guideline for protecting human health (20 μg/m³); only 16% of the study population has been previously exposed to levels exceeding 20 μg/m3 [13, 14]. However, in both emissions settings,

**Table 4. Descriptive statistics of $H_2S$ concentrations, stratified by outcomes.**

| | | $H_2S_{past}$[a] | | $H_2S_{current}$[b] | |
|---|---|---|---|---|---|
| | N | mean (SD) | p[c] | mean (SD) | p[c] |
| **COPD Stage I** | | | | | 0.194 |
| No | 1760 | 10.2 (10.4) | 0.179 | 0.80 (0.71) | |
| Yes | 255 | 11.2 (11.0) | | 0.86 (0.75) | |
| **COPD Stage II** | | | 0.139 | | |
| No | 1952 | 10.4 (10.42) | | 0.80 (0.72) | 0.087 |
| Yes | 63 | 9.8 (11.5) | | 0.73 (0.81) | |
| **FEV1%** | | | | | |
| FEV1 > = 80 | 1888 | 10.5 (10.5) | 0.011 | 0.81 (0.72) | 0.007 |
| FEV1 <80 | 127 | 8.4 (10.2) | | 0.67 (0.72) | |
| **FVC%** | | | | | |
| FVC > = 80 | 1942 | 10.51 (10.5) | 0.002 | 0.63 (0.89) | 0.002 |
| FVC <80 | 73 | 6.7 (8.5) | | 0.56 (0.60) | |

[a]$H_2S_{past}$: maximum average mobile concentration of $H_2S$ calculated over 90 days according to the past emissions scenario (six geothermal power plants' emissions: Bellavista, PC2, PC3, PC4; PC5, Bagnore 3)

[b] $H_2S_{current}$: maximum average mobile concentration of $H_2S$ calculated over 90 days according to the current emissions scenario (five geothermal power plants' emissions: PC3, PC4; PC5, Bagnore 3, Bagnore 4)

[c] p: p-values from Wilcoxon test

**Table 5. Percent variation (PV) and 95% confidence interval (95%CI) in FEV1, FVC and FEV1/FVC ratio for interquartile range increases in $H_2S$ concentration.**

|  | $H_2S_{past}$ [a] | | $H_2S_{current}$ [b] | |
|---|---|---|---|---|
|  | PV [c] | 95%CI | PV | 95%CI |
| **FEV1** | 1.10 | 0.23–1.9 | 0.99 | 0.17–1.80 |
| **FVC** | 1.66 | 0.88–2.45 | 1.48 | 0.476–2.21 |
| **FEV1/FVC** | -0.51 | -1.04–0.02 | -0.47 | -0.97–0.02 |

[a] $H_2S_{past}$: maximum average mobile concentration of $H_2S$ calculated over 90 days according to the past emissions scenario (six geothermal power plants' emissions: Bellavista, PC2, PC3, PC4; PC5, Bagnore 3)

[b] $H_2S_{current}$: maximum average mobile concentration of $H_2S$ calculated over 90 days according to the current emissions scenario (five geothermal power plants' emissions: PC3, PC4; PC5, Bagnore 3, Bagnore 4)

[c] PV: percent variation and 95%CI from multivariate generalized linear regression models associated with interquartile range $H_2S$ increase (13.8 µg/m³ for $H_2S_{pas}$, 3.5 µg/m³ for $H_2S_{current}$), adjusting for gender, age, education, BMI, pack-years, physical activity, mode of participation in InVETTA, cardiac comorbidities.

no evidence of an association with COPD was observed. In models using less complex exposure metrics, reduced lung function was not associated with living near a geothermal plant or the perception of $H_2S$ odour.

Our findings are consistent with those reported by another study conducted in Rotorua, New Zealand, a city hosting the world's largest community that lives in an active geothermal field [29]. Authors examined spirometric parameters associated with ambient $H_2S$ exposure and found no evidence of an association with COPD risk. In line with InVETTA results, the New Zealand study also highlighted negative associations, with evidence of better lung function in the higher exposure quartiles compared to the lowest exposure quartiles [29]. The absence of an association between respiratory outcome and exposure to $H_2S$ is also confirmed in studies conducted in other countries [30–33]. Conversely, other studies have shown an association between $H_2S$ exposure and a respiratory function deficit. Richardson has observed reduced FEV1/FVC levels in non-smoking sewer workers exposed to $H_2S$ [34]. However, this study has significant limitations due to the lack of specific data regarding exposure levels to hydrogen sulphide and the confounding effect of possible exposure to other toxic substances [34]. Kilburn [35] reported significant reductions in FEV1 and FVC in a sample of 25 persons residing near an intensive swine farm, compared with 58 persons from two reference groups. In addition to the low sample size, this study has a high risk of selection bias because the exposed group was self-organizing. Furthermore, two systematic reviews concerning the effects of low-level $H_2S$ exposure suggested no severe impact on the respiratory system and lung function [15, 16].

A possible explanation of the beneficial effect of $H_2S$ on lung function may be related to the essential role of endogenous $H_2S$ in cellular functions and physiological and pathological processes [36]. Recently, several studies have investigated the therapeutic potential of $H_2S$ for respiratory diseases, such as obstructive diseases, pulmonary fibrosis, emphysema, pancreatic inflammatory/respiratory lung injury, pulmonary inflammation, bronchial asthma and bronchiectasis [37–39]. Furthermore, in the respiratory tract, endogenous $H_2S$ has been shown to regulate important functions such as airway tone, pulmonary circulation, cell proliferation or apoptosis, fibrosis, oxidative stress and inflammation [40].

Our previous residential cohort study conducted in the Mt. Amiata area between 01/01/1998 and 31/12/2016 reported a 98% excess risk for COPD hospitalization (HR = 1.98 95%CI: 1.49–2.63) in subjects exposed to an $H_2S$ maximum 90-days moving average > 20 µg/m³ as compared with subjects exposed to <5 µg/m³ [20]. Although the same simulation modelling

**Table 6.** Associations between exposures (emissions, distance from geothermal plant and perceiving H₂S odour) and outcomes (COPD and lung function parameters) from multivariate logistic regression models.

| | $H_2S_{past}$ [a] | | $H_2S_{current}$ [b] | | Distance | | Odour perception | |
|---|---|---|---|---|---|---|---|---|
| | PR[c] | 95%CI | PR | 95%CI | PR | 95%CI | PR | 95%CI |
| **COPD Stage I** | | | | | | | | |
| All participants | 1.039 | 0.900–1.199 | 1.032 | 0.900–1.183 | 1.058 | 0.965–1.160 | 0.830 | 0.654–1.055 |
| Men | 1.064 | 0.875–1.293 | 1.046 | 0.872–1.255 | 1.044 | 0.922–1.181 | 0.760 | 0.543–1.064 |
| Women | 1.013 | 0.813–1.260 | 1.014 | 0.822–1.252 | 1.083 | 0.943–1.243 | 0.905 | 0.646–1.268 |
| 18–39 years | 0.873 | 0.442–1.721 | 0.763 | 0.394–1.476 | 1.416 | 1.050–1.911 | 0.747 | 0.308–1.808 |
| 40–54 years | 1.157 | 0.883–1.515 | 1.174 | 0.915–1.505 | 0.935 | 0.758–1.153 | 0.886 | 0.536–1.466 |
| 55–70 years | 1.028 | 0.866–1.220 | 1.016 | 0.846–1.197 | 1.072 | 0.963–1.194 | 0.837 | 0.633–1.106 |
| Main area | 1.114 | 0.954–1.301 | 0.110 | 0.958–1.287 | 0.907 | 0.748–1.101 | 0.894 | 0.680–1.177 |
| Control area | 1.429 | 0.503–4.060 | 1.391 | 0.546–3.542 | 0.951 | 0.769–1.176 | 0.763 | 0.441–1.321 |
| Sample | 0.994 | 0.841–1.176 | 0.986 | 0.839–1.157 | 1.087 | 0.979–1.208 | 0.762 | 0.560–1.038 |
| Volunteers | 1.119 | 0.866–1.445 | 1.123 | 0.881–1.432 | 0.952 | 0.793–1.143 | 0.894 | 0.609–1.314 |
| Never smokers | 1.166 | 0.901–1.510 | 1.122 | 0.886–1.420 | 1.087 | 0.909–1.300 | 0.638 | 0.415–0.979 |
| Smokers | 0.935 | 0.754–1.159 | 0.933 | 0.756–1.151 | 1.065 | 0.928–1.222 | 0.865 | 0.579–1.293 |
| **COPD Stage II** | | | | | | | | |
| All participants | 0.783 | 0.565–1.086 | 0.789 | 0.570–1.093 | 1.231 | 1.055–1.436 | 1.030 | 0.616–1.720 |
| Men | 0.844 | 0.567–1.302 | 0.838 | 0.538–1.305 | 1.167 | 0.904–1.506 | 1.009 | 0.449–2.268 |
| Women | 0.717 | 0.454–1.133 | 0.728 | 0.464–1.141 | 1.333 | 1.104–1.610 | 1.040 | 0.534–2.024 |
| 18–39 years | 0.192 | 0.026–1.407 | 0.166 | 0.019–1.450 | n.d.^ | n.d. | n.d. | n.d. |
| 40–54 years | 1.037 | 0.435–2.471 | 1.088 | 0.533–2.222 | 0.890 | 0.539–1.471 | n.d. | n.d. |
| 55–70 years | 0.793 | 0.554–1.134 | 0.790 | 0.552–1.130 | 1.250 | 1.051–1.487 | 0.905 | 0.522–1.571 |
| Main area | 0.860 | 0.602–1.228 | 0.876 | 0.615–1.249 | 1.247 | 0.860–1.807 | 1.235 | 0.639–2.386 |
| Control area | 0.717 | 0.118–4.343 | 0.602 | 0.095–3.795 | 1.074 | 0.741–1.556 | 0.993 | 0.412–2.390 |
| Sample | 0.831 | 0.599–1.153 | 0.848 | 0.612–1.174 | 1.197 | 0.994–1.440 | 0.851 | 0.462–1.570 |
| Volunteers | 0.538 | 0.228–1.271 | 0.513 | 0.230–1.144 | 1.317 | 0.998–1.739 | 1.783 | 0.549–5.785 |
| Never smokers | 0.602 | 0.277–1.310 | 0.622 | 0.289–1.341 | 1.510 | 1.105–2.602 | 0.944 | 0.305–2.924 |
| Smokers | 0.735 | 0.452–1.194 | 0.723 | 0.441–1.185 | 1.266 | 1.006–1.594 | 0.969 | 0.401–2.342 |
| **FEV1<80%** | | | | | | | | |
| All participants | 0.672 | 0.518–0.871 | 0.695 | 0.538–0.898 | 1.193 | 1.068–1.332 | 1.024 | 0.718–1.462 |
| Men | 0.735 | 0.480–1.125 | 0.751 | 0.495–1.140 | 1.080 | 0.891–1.310 | 1.115 | 0.583–2.133 |
| Women | 0.632 | 0.456–0.875 | 0.660 | 0.478–0.911 | 1.281 | 1.124–1.460 | 0.983 | 0.639–1.513 |
| 18–39 years | 0.512 | 0.224–1.170 | 0.587 | 0.264–1.307 | 1.299 | 0.965–1.750 | 1.328 | 0.449–3.927 |
| 40–54 years | 0.684 | 0.384–1.217 | 0.765 | 0.461–1.270 | 0.979 | 0.735–1.303 | 1.346 | 0.624–2.901 |
| 55–70 years | 0.718 | 0.529–0.976 | 0.713 | 0.527–0.966 | 1.254 | 1.095–1.436 | 0.885 | 0.578–1.355 |
| Main area | 0.759 | 0.581–0.991 | 0.789 | 0.606–1.028 | 1.140 | 0.880–1.476 | 1.377 | 0.862–2.199 |
| Control area | 0.385 | 0.083–1.780 | 0.414 | 0.106–1.605 | 1.025 | 0.786–1.338 | 0.703 | 0.364–1.357 |
| Sample | 0.728 | 0.552–0.961 | 0.760 | 0.578–0.999 | 1.198 | 1.044–1.375 | 0.815 | 0.514–1.291 |
| Volunteers | 0.478 | 0.277–0.824 | 0.477 | 0.288–0.790 | 1.191 | 0.981–1.447 | 1.559 | 0.800–3.037 |
| Never smokers | 0.861 | 0.564–1.316 | 0.910 | 0.613–1.350 | 1.246 | 1.004–1.546 | 1.221 | 0.625–2.387 |
| Smokers | 0.605 | 0.394–0.930 | 0.602 | 0.384–0.943 | 1.261 | 1.072–1.482 | 0.667 | 0.363–1.225 |
| **FVC<80%** | | | | | | | | |
| All participants | 0.483 | 0.316–0.739 | 0.522 | 0.348–0.781 | 1.151 | 0.987–1.343 | 1.105 | 0.676–1.806 |
| Men | 0.350 | 0.141–0.869 | 0.346 | 0.141–0.851 | 1.230 | 0.961–1.573 | 1.943 | 0.715–5.282 |
| Women | 0.543 | 0.335–0.881 | 0.607 | 0.386–0.955 | 1.104 | 0.901–1.352 | 0.868 | 0.481–1.568 |
| 18–39 years | 0.146 | 0.023–0.912 | 0.190 | 0.037–0.967 | 1.443 | 0.954–2.180 | 1.560 | 0.350–6.952 |
| 40–54 years | 0.705 | 0.389–1.278 | 0.759 | 0.0441–1.308 | 0.985 | 0.746–1.302 | 1.168 | 0.490–2.784 |

*(Continued)*

**Table 6.** (Continued)

| | $H_2S_{past}$ [a] | | $H_2S_{current}$ [b] | | Distance | | Odour perception | |
|---|---|---|---|---|---|---|---|---|
| | PR[c] | 95%CI | PR | 95%CI | PR | 95%CI | PR | 95%CI |
| 55–70 years | 0.422 | 0.223–0.796 | 0.455 | 0.254–0.816 | 1.200 | 0.961–1.499 | 1.010 | 0.533–1.912 |
| Main area | 0.524 | 0.341–0.803 | 0.561 | 0.371–0.849 | 1.210 | 0.858–1.705 | 1.265 | 0.697–2.295 |
| Control area | 0.228 | 0.017–2.969 | 0.300 | 0.035–2.537 | 0.868 | 0.572–1.319 | 1.045 | 0.358–3.044 |
| Sample | 0.502 | 0.293–0.860 | 0.542 | 0.326–0.904 | 1.141 | 0.927–1.406 | 0.823 | 0.398–1.702 |
| Volunteers | 0.426 | 0.224–0.809 | 0.454 | 0.250–0.823 | 1.212 | 0.956–1.537 | 1.383 | 0.635–3.014 |
| Never smokers | 0.754 | 0.402–1.141 | 0.808 | 0.445–1.469 | 1.152 | 0.854–1.554 | 1.295 | 0.592–2.834 |
| Smokers | 0.294 | 0.123–0.705 | 0.303 | 0.132–0.696 | 1.306 | 1.017–1.677 | 0.505 | 0.185–1.382 |

[a]$H_2S_{past}$: maximum average mobile concentration of $H_2S$ calculated over 90 days according to the past emissions scenario (six geothermal power plants' emissions: Bellavista, PC2, PC3, PC4; PC5, Bagnore 3)

[b] $H_2S_{current}$: maximum average mobile concentration of $H_2S$ calculated over 90 days according to the current emissions scenario (five geothermal power plants' emissions: PC3, PC4; PC5, Bagnore 3, Bagnore 4)

[c] PR: prevalence ratios and 95%CI from multivariate logistic regression models associated with interquartile range $H_2S$ increase (13.8 µg/m³ for $H_2S_{pas}$, 3.5 µg/m³ for $H_2S_{current}$), adjusting for gender, age, education, BMI, pack-years, physical activity, mode of participation in InVETTA, cardiac comorbidities; ^n.d: not determined

output for exposure assessment was used in the previous cohort study and the InVETTA survey, several factors may explain our different results. Firstly, the two populations are different in terms of age. In the previous cohort study, the population included all age groups, whereas, in InVETTA, the sample is 18 to 70 years old. Secondly, the InVETTA survey identified COPD cases through spirometric testing, whereas the previous cohort study was based on an administrative health database, and COPD cases were identified through hospital discharge diagnoses. For this reason, the two investigations may have included subjects at different levels of COPD severity. Indeed, in Italy, as in many other countries, managing various chronic diseases, such as COPD, has gradually shifted from a hospital-based approach to primary health care services. Thirdly, as opposed to the current InVETTA study, our previous cohort study did not include individual risk factors in multivariate models, limiting the ability to consider potential confounding.

Potential limitations of the present study, mainly in terms of selection bias, need to be considered. Among non-volunteers, the participation rate in the InVETTA survey was 46.1%, which is in line with the participation rates observed in recent epidemiological surveys. The steep decline in participation in scientific surveys is well documented, and over the last few years, many issues have been discussed to explain this declining trend [41, 42]. Some central reasons may include the increase in research studies, the proliferation in marketing surveys and political polls often indistinguishable from scientific surveys, the increasing complexity of research studies demanding long follow-up, biological samples and the growing popular disillusionment with science worldwide. The potential bias resulting from a low participation rate is a central issue in epidemiological research. Our study comparison analyses between respondents and non-respondents showed no significant differences.

In this study, we accepted volunteers to participate in the survey. This choice could have introduced a selection bias because volunteers could not represent the general population. However, comparisons between volunteers and participants randomly selected from municipal registries showed some differences: volunteers reported higher levels of education, lower percentages of smokers, they were less sedentary and overweight/obese. In addition, volunteers were less exposed to hydrogen sulphide than subjects randomly selected. For these reasons, we addressed this critical issue in the statistical protocol in various ways. First, we included in

multivariate models a dichotomous variable identifying the mode of participation (sample vs volunteers) as a potential confounder, and secondly, we conducted stratified analyses to remove any confounding. Although suffering from greater imprecision due to the lower population size, stratified analyses confirm the lack of association between exposure to hydrogen sulphide and respiratory health in both subgroups.

Regarding potential information bias, in our study, the assessment of $H_2S$ exposure was based on data from dispersion models produced according to international standards and simulation output validated using data from fixed monitoring sites. Past and current emission scenarios were included as input data in simulation models to consider both the effect of long-term exposure over the years and the effect of more recent exposure. Considering the residential history of InVETTA participants and the exposure levels in their work and study locations, including time spent at home or work, made it possible to maximize the accuracy of individual exposure estimation, minimizing the risk of introducing misclassification bias. Additional sensitivity analyses used the home distance from the nearest geothermal power plant and the odour perception as an exposure proxy. Distance is a crude approximation of exposure to ambient $H_2S$ concentrations since it does not consider meteorological conditions, which can strongly modify the exposure levels of the resident population. Odour perception is also a very subjective indicator. The pungent odour of $H_2S$ can cause a wide range of psychosomatic or even physiological symptoms, including respiratory complaints such as whistling and irritation of the nose or throat [15]. However, sensitivity analyses confirm results from primary analyses, i.e. the lack of association between increasing exposure to geothermal plants' emissions and adverse respiratory health effects. Another possible limitation was not performing spirometry testing post-bronchodilator; the choice was based on the nature of the study as it was a first-level screening survey.

Numerous risk factors and potential confounders were evaluated and included in multivariate models.

Individual information about lifestyle, habits, occupational exposure, home characteristics and clinical history were collected through a face-to-face administered questionnaire, according to the standard format used in similar cross-sectional surveys on respiratory health. Among environmental risk factors, we did not include other air pollutants typical of urban or industrial areas, such as particulate matter (PM) and nitrogen oxides ($NO_2$). In the Amiata area, spot surveys conducted by the Regional Agency for the Environmental Protection of Tuscany reported PM and $NO_2$ concentrations below the standard limit and WHO guidelines. Therefore, the potential role of these pollutants in determining the respiratory health of population is likely negligible. More relevant could be the role of arsenic and other metals, naturally present in local springs connected to the drinking water distribution system. Mt. Amiata is the second tallest volcano in Italy, and arsenic's widespread presence in groundwater is a well-known public health problem. For many years municipalities of the Mt. Amiata area requested derogations from the standard limit of As in drinking water (10 μg/l), until 2011, when huge intervention measures and structural works were put in place in order to restore water quality. An indicator of long-term exposure to arsenic in drinking water was calculated from water sampling campaigns conducted by the water utility company. Adjustment for this potential confounder did not modify our results on the relationship between respiratory outcome and $H_2S$ exposure.

## Conclusions

In conclusion, this large cross-sectional survey relies on objective and quality-controlled outcome measures, controlling for a wide set of potential confounders and comprehensive

modelling of long-term and current H$_2$S exposures. Its findings provide no evidence that chronic and current exposure to low levels of H$_2$S is associated with impairment of pulmonary function. Instead, they suggest that ambient H$_2$S exposure at levels measured in Mt. Amiata may benefit lung function.

## Supporting information

**S1 Checklist. STROBE statement—Checklist of items that should be included in reports of observational studies.**
(DOCX)

## Acknowledgments

We would like to thank the health professionals and administrative personnel whose valuable work was essential for the implementation of the survey.

## Author Contributions

**Conceptualization:** Daniela Nuvolone, Fabio Voller.

**Data curation:** Davide Petri.

**Formal analysis:** Giorgia Stoppa, Daniela Nuvolone, Davide Petri.

**Funding acquisition:** Fabio Voller.

**Investigation:** Giorgia Stoppa, Daniela Nuvolone, Davide Petri, Emanuele Crocetti, Fabio Barbone.

**Methodology:** Giorgia Stoppa, Daniela Nuvolone, Emanuele Crocetti, Fabio Barbone, Fabio Voller.

**Supervision:** Daniela Nuvolone, Fabio Voller.

**Writing – original draft:** Giorgia Stoppa, Daniela Nuvolone.

**Writing – review & editing:** Davide Petri, Letizia Centi, Francesca Nisticò, Emanuele Crocetti, Fabio Barbone, Fabio Voller.

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
