## [Decision Letter · Decision Letter 0]

5 Sep 2023

PONE-D-23-15844Exposure to low levels of hydrogen sulphide and its impact on chronic obstructive pulmonary disease and lung function in the geothermal area of Mt. Amiata in Italy: the cross-sectional InVETTA study.PLOS ONE

Dear Dr. Stoppa,

Thank you for submitting your manuscript to PLOS ONE. After careful consideration, we feel that it has merit but does not fully meet PLOS ONE’s publication criteria as it currently stands. Therefore, we invite you to submit a revised version of the manuscript that addresses the points raised during the review process.

We look forward to receiving your revised manuscript.

Kind regards,

George Kuryan

Academic Editor

PLOS ONE

Journal Requirements:

3. We note that Figure 1 in your submission contain map images which may be copyrighted. All PLOS content is published under the Creative Commons Attribution License (CC BY 4.0), which means that the manuscript, images, and Supporting Information files will be freely available online, and any third party is permitted to access, download, copy, distribute, and use these materials in any way, even commercially, with proper attribution. For these reasons, we cannot publish previously copyrighted maps or satellite images created using proprietary data, such as Google software (Google Maps, Street View, and Earth). For more information, see our copyright guidelines: http://journals.plos.org/plosone/s/licenses-and-copyright.

Reviewers' comments:

Reviewer's Responses to Questions

**Comments to the Author**

1. Is the manuscript technically sound, and do the data support the conclusions?

Reviewer #1: Yes

2. Has the statistical analysis been performed appropriately and rigorously? 

Reviewer #1: I Don't Know

3. Have the authors made all data underlying the findings in their manuscript fully available?

Reviewer #1: Yes

4. Is the manuscript presented in an intelligible fashion and written in standard English?

Reviewer #1: Yes

5. Review Comments to the Author

Reviewer #1: Good analysis of data on the effect of H2S exposure on on lung function and prevalence of chronic obstructive pulmonary disease. There are few clarifications needed which are highlighted as comments in the attached manuscript.

6. PLOS authors have the option to publish the peer review history of their article (what does this mean?). If published, this will include your full peer review and any attached files.

Reviewer #1: **Yes: **Balamugesh Thangakunam

---

## [Author Response · Author response to Decision Letter 0]

6 Oct 2023

I am writing to express my sincere gratitude for your invaluable and comprehensive review of my submission. . Your detailed feedback and meticulous examination of the manuscript were immensely helpful.

Your perceptive comments and insightful suggestions, presented point by point in the attached file, have been instrumental in enhancing the quality and rigor of my work.

---

## [Editor Report · Decision Letter 1]

17 Oct 2023

Exposure to low levels of hydrogen sulphide and its impact on chronic obstructive pulmonary disease and lung function in the geothermal area of Mt. Amiata in Italy: the cross-sectional InVETTA study.

PONE-D-23-15844R1

Dear Dr. Stoppa,

We’re pleased to inform you that your manuscript has been judged scientifically suitable for publication and will be formally accepted for publication once it meets all outstanding technical requirements.

Kind regards,

George Kuryan

Academic Editor

PLOS ONE
---

## [Editor Report · Acceptance letter]

23 Oct 2023

PONE-D-23-15844R1 

Exposure to low levels of hydrogen sulphide and its impact on chronic obstructive pulmonary disease and lung function in the geothermal area of Mt. Amiata in Italy: the cross-sectional InVETTA study. 

Dear Dr. Stoppa:

I'm pleased to inform you that your manuscript has been deemed suitable for publication in PLOS ONE. Congratulations! Your manuscript is now with our production department. 

Kind regards, 

on behalf of

Professor George Kuryan 

Academic Editor

PLOS ONE